# Antagonistic Activity of *Streptomyces alfalfae* 11F against *Fusarium* Wilt of Watermelon and Transcriptome Analysis Provides Insights into the Synthesis of Phenazine-1-Carboxamide

**DOI:** 10.3390/plants12223796

**Published:** 2023-11-08

**Authors:** Dan Dong, Maoying Li, Taotao Zhang, Zhenfeng Niu, Guoping Xue, Hongmei Bai, Wenyu Zhao, Jiajia Yu, Wei Jiang, Huiling Wu

**Affiliations:** 1Institute of Plant Protection, Beijing Academy of Agriculture and Forestry Sciences, Key Laboratory of Environment Friendly Management on Fruit and Vegetable Pests in North China (Co-Construction by Ministry and Province), Ministry of Agriculture and Rural Affairs, Beijing 100097, China; dan20080801@163.com (D.D.); ztt1024@163.com (T.Z.); nzf1750@163.com (Z.N.); 17801008226@163.com (W.Z.); 18437977523@163.com (J.Y.); 2National Watermelon and Melon Improvement Center, Beijing Academy of Agriculture and Forestry Sciences, Beijing 100097, China; limaoying@nercv.org; 3Inner Mongolia Academy of Agricultural and Animal Husbandry Sciences, Hohhot 010031, China; 18048320325@163.com (G.X.); 15848909291@163.com (H.B.)

**Keywords:** biological control, *Streptomyces alfalfae*, differentially expressed genes, secondary metabolites, RNA-seq

## Abstract

*Streptomyces alfalfa* strain 11F has inhibitory effects on many phytopathogenic fungi and improves the establishment and biomass yield of switchgrass. However, the antagonistic effects of strain 11F on *Fusarium* wilt of watermelon and its secondary metabolites that contribute to its biocontrol activity are poorly understood. We evaluated the antagonistic and growth-promoting effects of strain 11F and conducted a transcriptome analysis to identify the metabolites contributing to antifungal activity. Strain 11F had marked inhibitory effects on six fungal pathogens. The incidence of *Fusarium* wilt of watermelon seedlings was decreased by 46.02%, while watermelon seedling growth was promoted, as indicated by plant height (8.7%), fresh weight (23.1%), and dry weight (60.0%). Clean RNA-sequencing data were annotated with 7553 functional genes. The 2582 differentially expressed genes (DEGs) detected in the Control vs. Case 2 comparison were divided into 42 subcategories of the biological process, cellular component, and molecular function Gene Ontology categories. Seven hundred and forty functional genes (55.47% of the DEGs) were assigned to Kyoto Encyclopedia of Genes and Genomes metabolic pathways, reflecting the complexity of the strain 11F metabolic regulatory system. The expression level of the gene *phzF*, which encodes an enzyme essential for phenazine-1-carboxylic acid (PCA) synthesis, was downregulated 3.7-fold between the 24 h and 48 h fermentation time points, suggesting that strain 11F can produce phenazine compounds. A phenazine compound from 11F was isolated and identified as phenazine-1-carboxamide (PCN), which contributed to the antagonistic activity against *Fusarium oxysporum* f. sp. *niveum*. PCA was speculated to be the synthetic precursor of PCN. The downregulation in *phzF* expression might be associated with the decrease in PCA accumulation and the increase in PCN synthesis in strain 11F from 24 to 48 h. *Streptomyces alfalfae* 11F protects watermelon seedlings from *Fusarium* wilt of watermelon and promotes seedling growth. The transcriptome analysis of strain 11F provides insights into the synthesis of PCN, which has antifungal activity against *F. oxysporum* f. sp. *niveum* of watermelon.

## 1. Introduction

The large-scale use of traditional chemical pesticides has resulted in serious environmental pollution. The increase in prevalence of pesticide residues and bacterial resistance has motivated the use of environmentally friendly and sustainable microbial control methods. Recently, antibiotics, stress inducers, and biocontrol microorganisms have been largely used for the prevention of plant diseases. The main biocontrol microorganisms are bacteria, actinomycetes, fungi, and viruses [1], which can either secrete antimicrobial substances or perform niche competition to promote plant growth and induce plant systemic resistance [2]. Microbial biological control agents are effective alternatives to synthetic chemicals [3]. In the present study, *Streptomyces alfalfae* strain 11F isolated from the soil near Qinghai Lake (101°77′ E, 36°62′ N) in China was observed to have inhibitory effects on the phytopathogenic fungi *Fusarium oxysporum* f. sp. *niveum*, *F. solani*, *F. graminearum*, *Setosphaeria turcica*, *Botryosphaeria dothidea*, and *Botrytis cinerea*. Strain 11F can produce indole-3-acetic acid and siderophores and is known to have phosphate-solubilizing and N_2_-fixing abilities. In addition, strain 11F improves the establishment and biomass yield of switchgrass [4]. With analyses of phylogenetic trees constructed on the basis of 16S rRNA sequences and genomic collinearity using Mauve_installer_2.3.1, strain 11F has been shown to have a conserved genome structure and collinear relationships to *S. alfalfae* ACCC40021 [4]. In China, *S. alfalfae* ACCC40021, a rhizobacterium beneficial to plants, is widely used as a microbial fertilizer against phytopathogens [5]. However, little information is available regarding its antibacterial active substances and related functional genes.

In this study, we evaluated the effects of *S. alfalfae* 11F on *Fusarium* wilt resistance and on the promotion of seedling growth in watermelon and analyzed the differentially expressed genes (DEGs) induced by the fermentation of strain 11F. The results clarified that the 11F strain could promote plant growth and enhance disease resistance in watermelon and identified the antifungal active substances PCA and PCN and its functional gene *phzF*, contributing to the biocontrol activities of strain 11F.

## 2. Results

### 2.1. Antimicrobial Activity Assay

Strain 11F had strongly antagonistic effects on all tested phytopathogenic fungi, especially *F. oxysporum* f. sp. *niveum* and *B. dothidea* (Appendix A). After incubation of the dual-culture plates for 10 days, the mean diameter of *Fusarium oxysporum* f. sp. *niveum* colonies co-cultured with strain 11F was 3.1 ± 0.2 cm, which was significantly smaller than the diameter of colonies in the absence of strain 11F (7.8 ± 0.31 cm) (Figure 1A,B). The diameter of the growth inhibition zones of the 48 h fermentation broth (1.6 ± 0.15 cm) was significantly larger than that of the 24 h fermentation broth (1.0 ± 0.22 cm) (Figure 1C). Moreover, strain 11F induced morphological changes to *F. oxysporum* f. sp. *niveum* mycelia, which were curled and excessively branched. In addition, the mycelial tips expanded to form a spherical structure (Figure 2A,B), which was in contrast to the morphology of the control mycelium (Figure 2C,D). The direct confrontation assay revealed the significant antagonistic effect of strain 11F on *F. oxysporum* f. sp. *niveum* (42.3% inhibition rate).

### 2.2. Strain 11F Enhanced Watermelon Plant Growth and Disease Resistance

The effects of strain 11F on watermelon seedling characteristics and the development of *Fusarium* wilt were evaluated. Treatment with strain 11F significantly increased watermelon seedling height, fresh weight, and dry weight. Compared with those of the control, the height, fresh weight, and dry weight of the watermelon seedlings treated with 60× diluted fermentation broth (6 × 10^6^ cfu/mL) increased by 8.7%, 23.1%, and 60.0%, respectively (Figure 3 and Table 1).

Strain 11F significantly decreased the severity of the disease caused by *F. oxysporum* f. sp. *niveum* on watermelon plants. The disease severity index (DSI) and incidence rate of the *Fusarium*-inoculated plants treated with strain 11F decreased to 27.57% and 25.14%, respectively, which were significantly lower than the DSI (73.59%) and incidence rate (100%) of the *Fusarium*-inoculated plants without strain 11F (Table 2). Thus, strain 11F protected the watermelon plants against infection by *F. oxysporum* f. sp. *niveum* and significantly enhanced watermelon plant growth.

### 2.3. Transcriptome Sequencing, Data Quality, and Transcript Assembly

The transcriptome of strain 11F was sampled and sequenced at three fermentation time points: 6 h (Control), 24 h (Case 1), and 48 h (Case 2). The clean read percentage of each sample exceeded 95% and the GC content ranged from 58% to 68% (Figure 4A). The percentage of bases with Q20 (high sequencing quality) approached 99% (Figure 4B and Appendix A). The mapping results obtained using Bowtie2 (https://bowtie-bio.sourceforge.net/bowtie2/manual.shtml, accessed on 17 September 2023) software are provided in the Appendix A. For the three sampling time points, the average numbers of mapped reads (i.e., effective mRNA information) were 14,537,664 (97.31%), 13,940,012 (94.32%), and 13,400,250 (90.92%), and the multiple comparison percentage was ≤10% (Appendix A). Accordingly, the quality of the transcriptome sequencing data was sufficient for further analyses.

### 2.4. Differential Gene Expression Analysis

A total of 181 DEGs were common to all three comparisons of the RNA-seq data (Figure 5A), comprising 129 upregulated genes and 52 downregulated genes. The Control vs. Case 1 comparison revealed 1873 DEGs, consisting of 1056 upregulated genes and 817 downregulated genes. The Case 1 vs. Case 2 comparison detected 948 DEGs, of which 458 were upregulated genes and 490 were downregulated genes. A total of 2582 DEGs were identified by the Control vs. Case 2 comparison, comprising 1259 upregulated genes and 1323 downregulated genes (Figure 5B). Compared with that for the Control vs. Case 1 comparison, the number of DEGs for the comparison of Case 1 vs. Case 2 was reduced by 49.38%. In addition, enrichment plots revealed distinct changes to the highly enriched DEGs during the fermentation period. From Case 1 to Case 2, the number of DEGs associated with cellular components decreased, whereas the number of DEGs associated with biological processes increased (Figure 5C,D). The expression patterns of DEGs between Case 1 and Case 2 are shown in a cluster analysis heatmap (Appendix A).

### 2.5. Verification of DEG Expression Levels by Quantitative Real-Time PCR Analysis

To verify the changes in expression levels of the DEGs detected by the fragments per kilobase of transcript per million fragments mapped (FPKM) analysis, 19 DEGs were selected for a quantitative real-time PCR (qRT-PCR) analysis (Figure 6 and Appendix A). The expression levels of 14 genes were upregulated by more than 4-fold. These genes encoded the following proteins: phosphate ABC transporter substrate-binding protein PstS, DegT/DnrJ/EryC1/StrS aminotransferase, GNAT family N-acetyltransferase, phosphate ABC transporter ATP-binding protein, N-acetylmuramoyl-L-alanine amidase, aminoglycoside phosphotransferase family protein, phosphate ABC transporter permease PtsA, AraC family transcriptional regulator, ergothioneine biosynthesis protein EgtC, amino acid acetyltransferase, ABC transporter ATP-binding protein, FadR family transcriptional regulator, and polysaccharide biosynthesis protein. The expression levels of the remaining five genes were downregulated by more than 4-fold. These genes encoded the following proteins: TerD family protein, MFS transporter, ABC transporter substrate-binding protein, nitrate reductase, and fructose-bisphosphate aldolase. The qRT-PCR data were generally consistent with the results of the FPKM analysis and thus were indicative of the reliability of the RNA-seq data.

### 2.6. Functional Annotation and Enrichment Analysis of the DEGs

All DEGs were included in the gene ontology (GO) analysis. The DEGs in the Control vs. Case 2 comparison were classified into three main functional groups, comprising 18 biological process subcategories, 12 molecular function subcategories, and 12 cellular component subcategories (Figure 7). More specifically, “metabolic process” (920 DEGs), “cellular process” (933 DEGs), and “single biological processes” (706 DEGs) were the most common GO terms in the biological process category. Regarding the cellular component category, the predominant terms were “cell” (794 DEGs), “membrane part” (414 DEGs), and “cell part” (790 DEGs). In the molecular function category, the most commonly assigned terms were “catalytic activity” (1004 DEGs) and “binding” (861 DEGs). These findings reflect the enrichment of the biological process and molecular function categories among the DEGs.

The Kyoto Encyclopedia of Genes and Genomes (KEGG) pathway database is useful for systematic analyses of the metabolic pathways and functions of gene products in cells. Such analyses provide information regarding molecular interaction networks and the unique changes to individual biological pathways in organisms. The present KEGG pathway analysis indicated that the DEGs in the Control vs. Case 2 comparison participate in 180 pathways in six major categories and 29 subcategories. Moreover, 740 functional genes, accounting for 55.47% of the DEGs, were assigned to KEGG metabolic pathways. The ribosome (ko03010) and acarbose and validamycin biosynthesis (ko00525) pathways were significantly enriched (*p* < 0.05) among the DEGs in the Control vs. Case 2 comparison. In the Control vs. Case 1 comparison, the ribosome (ko03010) and porphyrin and chlorophyll metabolism (ko00860) pathways were significantly enriched (*p* < 0.05) among the DEGs. The significantly enriched (*p* < 0.05) pathways among the DEGs in the Case 1 vs. Case 2 comparison were ABC transporter (ko02010), acarbose and validamycin biosynthesis (ko00525), MAPK signaling pathway (ko04011), oxidative phosphorylation (ko00190), polyketide sugar unit biosynthesis (ko00523), longevity regulating pathway (ko04211\ko04212\ko04213), sulfur metabolism (ko00920), and arginine biosynthesis (ko00220) (Appendix A). Metabolic pathways were mainly enriched from 24 h to 48 h, which was the period in which strain 11F metabolic pathways were most active. Notably, the expression levels of eight DEGs involved in 10 steps of the acarbose and validamycin biosynthesis pathway (i.e., those involving *acbR*, *valC*, *acbC*, *valK*, *acbK*, *acbS*, *acbU*, *acbV*, *rfbA*, and *rfbB*) were upregulated between the 24 h and 48 h fermentation time points (Figure 8).

### 2.7. Expression of phzF and Characterization of a Phenazine Compound in Strain 11F

Among the DEGs, the expression level of *phzF* was downregulated 3.7-fold between the 24 h and 48 h fermentation time points. PhzF is an isomerase that catalyzes *trans*-2,3-dihydro-3-hydroxyan-thranilic acid isomerization and plays an essential role in the phenazine biosynthetic pathway in *Pseudomonas* species. Phenazine and its derivatives are nitrogen-containing heterocyclic redox agents with broad-spectrum activity against gram-positive and gram-negative bacteria [6], fungi, and algae [7], and these compounds are produced mainly by *Pseudomonas* and *Streptomyces* species [8]. In the present study, we isolated the phenazine compound that had antifungal activity against *F. oxysporum* f. sp. *niveum* in accordance with the differential expression of *phzF*. The molecular formula of the phenazine compound is C_13_H_9_N_3_O, ESI-MS *m*/*z*: 224.08 [M + H]^+^, 246.06 [M + Na]^+^; ^1^H NMR (500 MHz, DMSO-d6) *δ*: 9.72 (1H, s, NH2), 8.69 (1H, dd, *J* = 7.0 Hz, 1.7 Hz, H-2), 8.44 (1H, dd, *J* = 8.7 Hz, 1.7 Hz, H-4), 8.42 (1H, m, H-6), 8.31 (1H, m, H-9), 8.09 (^1^H, s, NH2), 8.07 (1H, m, H-3), 8.06 (1H, m, H-7), 8.04 (1H, m, H-8); ^13^C NMR (Table 3) (125 MHz, DMSO-d6) *δ*: 165.7 (CONH2), 142.8 (C-4a), 142.6 (C-9a), 141.3 (C-5a), 140.2 (C-10a), 134.1 (C-2), 133.0 (C-4), 132.0 (C-7), 131.6 (C-8), 130.9 (C-1), 130.3 (C-3), 129.3 (C-6), 129.2 (C-9). The structure of the phenazine compound was confirmed by heteronuclear multiple bond correlation (HMBC) and ^1^H-^1^H correlation spectroscopy (^1^H-^1^HCOSY) experiments (Figure 9A and Appendix A) and comparison of the spectra with reference spectra [9,10]. The phenazine compound was elucidated as phenazine-1-carboxamide (PCN; Figure 9B) and its antagonistic activity against *F. oxysporum* f. sp. *niveum* was verified by an inhibition zone test (Figure 9C).

## 3. Discussion

*Streptomyces*-derived compounds have potent bioactivities for pharmaceutical applications, including antibacterial, antifungal, anticancer, antitumor, cytotoxic, cytostatic, anti-inflammatory, antiparasitic, antimalarial, antiviral, antioxidant, and anti-angiogenesis activities [11,12]. In the present study, strain 11F decreased the disease symptoms in watermelon and altered *F. oxysporum* f. sp. *niveum* mycelial morphology. Secondary metabolites of strain 11F affected *F. oxysporum* f. sp. *niveum* mycelial differentiation. Takemoto et al. demonstrated that diphenylene iodonium, which is an NADPH oxidase inhibitor, decreases wild-type *Epichloë festucae* colony size and induces hyphal hyperbranched [13]. Similar results were obtained during an investigation of the antagonistic effects of *Bacillus subtilis* SG6 on *F. graminearum* D187 [14]. However, the mechanisms mediating the changes to fungal mycelial morphology require more thorough exploration. It is possible that strain 11F-induced stress changes the mycelial cell wall composition of *F. oxysporum* f. sp. *niveum*.

Transcriptome analyses are not only useful for clarifying transcript-level changes in developing cells and during responses to environmental perturbations, but may also contribute to the elucidation of unknown regulatory patterns. Transcriptome analyses have been conducted to comprehensively study the transcripts of model organisms, including *Arabidopsis thaliana*, *Drosophila melanogaster*, *Saccharomyces cerevisiae*, and *Oryza sativa* [14,15,16]. In the current study, the antifungal activities of the strain 11F fermentation broth against *F. oxysporum* f. sp. *niveum* varied among the examined fermentation time points. Regarding *Streptomyces* growth and development, the highly regulated synthesis of secondary metabolites generally begins after the morphological differentiation process is initiated [17,18,19]. Therefore, strain 11F gene expression in different fermentation periods was analyzed to screen for DEGs associated with secondary metabolite synthesis. Validamycin A, which is a trehalase inhibitor, has been used to prevent fungal infections of agricultural products [20]. The present results revealed that the acarbose and validamycin synthesis pathways are significantly enriched. Eight DEGs (*acbR*, *valC*, *acbC*, *valK*, *acbK*, *acbS*, *rfbA*, and *rfbB*) involved in 10 steps of these pathways were much more highly expressed at 48 h than at 24 h. In addition, the ABC transporter directly involved in antibiotic synthesis is significantly enriched. Many antibiotic-producing actinomycetes produce at least one ABC transporter (ATP-binding cassette) that participates in the antibiotic biosynthesis pathway and mediates the drug resistance of heterologous hosts [21]. Thus, ABC transporters may be responsible for the secretion of antibiotics from strain 11F cells.

High-throughput sequencing technology has been applied to reveal expression patterns in transcriptomes, identify novel genes and molecular markers, and clarify the regulation of non-coding RNA [22]. The technology has been widely used for the identification and functional characterization of genes in plants and microorganisms [23]. Recent research has clarified the transcriptional changes during bacterial–fungal/bacterial interactions. For example, Hennessy et al. demonstrated that the expression levels of genes involved in metabolite detoxification are highly upregulated in *Pseudomonas fluorescens* In5 co-cultured with plant pathogens, especially the fungus *Rhizoctonia solani* [24]. Li et al. characterized reference genes differentially expressed in *S. coelicolor* [25]. Arseneault et al. reported that the production of PCA by the biocontrol agent *P. fluorescens* LBUM223 affects several virulence-related cellular processes in *S. scabies*, including mycelial formation and oxidation–reduction homeostasis, and alters the expression of several well-characterized genes [26]. We found that strain 11F produced phenazine compounds, in part based on the differential expression of *phzF* with the analysis of DEGs. In addition, the antifungal metabolite PCN was isolated and identified, in accordance with the characteristics of phenazine compounds, and contributed to the biocontrol activity against *Fusarium* wilt in watermelon.

It is known that PhzF is an enzyme essential for PCA synthesis in *Pseudomonas* species. The production of PCN, rather than PCA, is the crucial metabolite required for the biocontrol ability of strain PCL1391 in the tomato–*F. oxysporum* test system [27]. There are few reports on the function of *phzF* in phenazine biosynthesis in *Streptomyces* species. PhzF is an essential enzyme for PCA synthesis in *Pseudomonas* spp.; the downregulation of its expression may be associated with the decrease in PCA accumulation and the increase in PCN synthesis in strain 11F from 24 to 48 h. However, we failed to identify the phenothiazine compound biosynthetic gene cluster in a genome-wide scan using the antiSMASH database (Appendix A). It is possible that *phzF* may perform a completely different regulatory role in phenazine synthesis in strain 11F. In future research, genome mining from genomic bacterial artificial chromosome (BAC) libraries, using the LEXAS system [28], will be conducted to mine gene clusters associated with phenazine compound biosynthesis in strain 11F. Synthesis of PCN will guide the optimization of fermentation processes in strain 11F and provide a strong basis for the further precise application of this strain in agriculture.

## 4. Materials and Methods

### 4.1. Microbe Strains and Culture Conditions

*Streptomyces alfalfae* strain 11F was cultured at 28 °C on potato dextrose agar (PDA) slants to induce spore formation [29]. The spores were used to inoculate a 50 mL seed culture medium (15 g/L sucrose, 40 g/L peptone, 0.5 g/L K_2_HPO_4,_ 0.5 g/L CaCO_3_, 0.5 g/L NaCl, and 0.5 g/L MgSO_4_^.^7H_2_O, pH 6.0). The seed cultures were incubated at 28 °C for 36 h on a rotary shaker (200 rpm). A yeast–salt–glycerol (ISP2: 4 g/L glucose, 4 g/L yeast extract, and 10 g/L malt extract, pH 7.2) [30] medium was used for the production of antifungal substances. The following plant pathogens were included in this study: *F. oxysporum* f. sp. *niveum*, *S. turcica*, *F. solani*, *B. dothidea*, *F. graminearum*, and *B. cinerea*. All of them were preserved at the Institute of Plant Protection, Beijing Academy of Agriculture and Forestry Sciences, China.

### 4.2. Assay of the S. alfalfae 11F Antimicrobial Spectrum

The antimicrobial spectrum of strain 11F was determined in accordance with a slightly modified dual-culture assay [31,32]. Hyphal plugs (7 mm diameter) were excised from a 3-day-old fungal culture on PDA. Each plug was placed at the center of a PDA plate, which was then incubated at 25 °C for 1 day. On either side of the plug, the PDA was inoculated with a single streak of strain 11F from PDA slants (2 cm from the plug). The plate was further incubated in darkness at 28 °C for 3 days. The antagonistic effects of strain 11F on *F. oxysporum* f. sp. *niveum* mycelial growth were examined. Plates containing PDA that were not inoculated with bacteria were used as controls. The assay was performed using three plates. The percentage inhibition (I%) was calculated as the following: I% = [(*R* − *r*)]/*R* × 100, where the radial growth of each *Fusarium* colony was measured in the presence (*r*) and absence (*R*) of strain 11F. The edges of the colonies cultured for 3 days were removed using a sterile scalpel and examined (×400 magnification) using a DMIRE2 light microscope (Leica, Wetzlar, Germany). All analyses were repeated three times.

### 4.3. Pot Experiment

A pot experiment was conducted in an air-conditioned greenhouse at the Institute of Plant Protection, Beijing Academy of Agriculture and Forestry Sciences, China. Sterilized watermelon seeds were germinated and then seedlings of uniform growth at the 3- to 4-leaf stages were selected for the following four treatments: (1) seedlings were treated with strain 11F (1 × 10^7^ cfu/mL) and then inoculated with a pathogen conidial suspension (5 × 10^6^ cfu/mL) for 15 min; (2) seedlings were only inoculated with a pathogen conidial suspension for 15 min; (3) seedlings were only treated with strain 11F; or (4) seedlings were treated with sterile distilled water. Twelve replicates of experimental pots were arranged in a randomized block design. The DSI and biocontrol effect were assessed when 50% of the plants were diseased at 10 days post-inoculation in the greenhouse. Growth characteristics such as plant height, number of tillers per plant, leaf size, and plant biomass, with or without treatment with strain 11F, were analyzed after 1 month to assess the effects of strain 11F on seedling growth.

In this experiment, the classification of the severity of the disease and the calculation method of the disease of flue cured watermelon wilt was strictly in accordance with the Survey method of the watermelon wilt plant diseases classification to evaluate the disease severity classification [33,34].

The disease severity index (DSI) was calculated as the following: ∑((disease progression • number of plants in this class)/(number of the highest incidence • number of plants investigated)).

### 4.4. Sample Collection and Library Construction

Total RNA was extracted from *S. alfalfae* strain 11F mycelia collected at three fermentation time points (6 h for the Control, 24 h for Case 1, and 48 h for Case 2). Mycelia were harvested by centrifugation at 12,000× *g* for 10 min at 4 °C and then immediately frozen in liquid nitrogen before being ground to powder using a mortar and pestle. Total RNA was extracted using the TRIzol reagent (Invitrogen, Waltham, MA, USA). The following nine samples were obtained (three replicates for each of the three time points): 171109D_6_1, 171109D_6_2, 171109D_6_3, 171109D_24_1, 171109D_24_2, 171109D_24_3, 171109D_48_1, 171109D_48_2, and 171109D_48_3. The RNA concentration was determined using a NanoDrop spectrophotometer (Thermo Fisher Scientific, Shanghai, China) and RNA quality was assessed using an Agilent Bioanalyzer (Agilent Technologies Inc., Santa Clara, CA, USA). The high-quality RNA was used to construct cDNA libraries.

### 4.5. Illumina HiSeq Sequencing and Analysis

The cDNA libraries constructed for the nine RNA samples were sequenced using an Illumina HiSeq 2000 sequencing platform by Beijing Boao Jing Dian Biotechnology Co., Ltd. (Beijing, China). The raw data were processed by eliminating the low-quality sequences to obtain high-quality clean data. After a stringent quality control step, the sequencing coverage and depth were determined using TopHat2 [35]. High-quality sequences were generated according to the reference genome (GCF_001975025.1_ASM197502v1_genomic.fna downloaded from NCBI). RSEM-1.2.26 [36] was used for quantitative analyses. Genes were annotated on the basis of the Pfam database and the reference genome (GCF_001975025.1_ASM197502v1_genomic.gtf downloaded from NCBI). The DEGs were identified using the DESeq2 package [37] in R (version 3.4.2) and the following criteria: |log_2_ (fold-change)| ≥ 1 and *p* < 0.05. Genes with an adjusted *p*-value less than 5% (according to the false discovery rate) were considered to be differentially expressed. The DEGs were functionally annotated using the BLASTP algorithm and the Gene Ontology (GO) and Kyoto Encyclopedia of Genes and Genomes (KEGG) databases.

### 4.6. Quantitative Real-Time PCR Analysis of DEGs

The identified DEGs were verified by a qRT-PCR analysis. Total RNA (1 µg) was used to synthesize cDNA for the qRT-PCR, which was performed using SYBR^®^ Premix Ex Taq™ II (Tli RNaseH Plus) (Takara, Beijing, China), supplemented with ROX and the CFX96™ Optics Module Real-Time PCR System (BIO−RAD, Hercules, CA, USA). The qRT-PCR primers were designed according to the reference gene sequences using SnapGene 5.3 software. Relative expression levels were determined using the 2^−ΔΔ*C*t^ method [38].

### 4.7. Isolation, Purification, and Determination of the Antifungal Metabolite

The fermentation broth of strain 11F was centrifuged at 12,000× *g* for 10 min to remove the cellular debris and the supernatant was extracted in an equal volume of ethyl acetate. The organic layer was concentrated under reduced pressure and the crude extract was dissolved in methanol.

The crude extract was chromatographed by thin-layer chromatography using a chloroform–methanol mixture (9:1) as the eluent and semipreparative reversed-phase high-performance liquid chromatography. The structure of the compound was elucidated by an analysis of spectroscopic and spectrometric data (1D, 2D-NMR, and HRESIMS). Determination of the structure of the compound was confirmed by HMBC and ^1^H-^1^HCOSY experiments.

### 4.8. Data Analysis

Raw data were organized using Excel 2010 and analyzed using IBM SPSS Statistics 22.0 software. The significance of differences among treatments was evaluated using Duncan’s multiple range test (*p* < 0.05). The results were expressed as means ± standard error (SE).

## 5. Conclusions

In this study, *S. alfalfae* strain 11F was highly antagonistic toward fungal pathogens tested. The greenhouse experiment indicated that the fermentation broth of strain 11F protected watermelon seedlings from *Fusarium* wilt to a certain extent and promoted seedling growth. Following transcriptome sequencing, the number of DEGs and enriched pathways changed significantly during the fermentation period. The number of DEGs involved in biological processes increased with prolonged fermentation from 6 h to 48 h. The GO analysis revealed that the biological process and molecular function categories were enriched among the DEGs, whereas the KEGG pathway analysis suggested that metabolic pathways were most active in strain 11F from 24 h to 48 h. The qRT-PCR analysis confirmed the reliability of the RNA-seq data. The phenazine substance PCN, which has antifungal activity against *F. oxysporum* f. sp. *niveum* based on the differential expression of *phzF*, was isolated. Further research will include the use of the LEXAS system to mine gene clusters responsible for phenazine compound biosynthesis in strain 11F.

## Figures and Tables

**Figure 1 plants-12-03796-f001:**
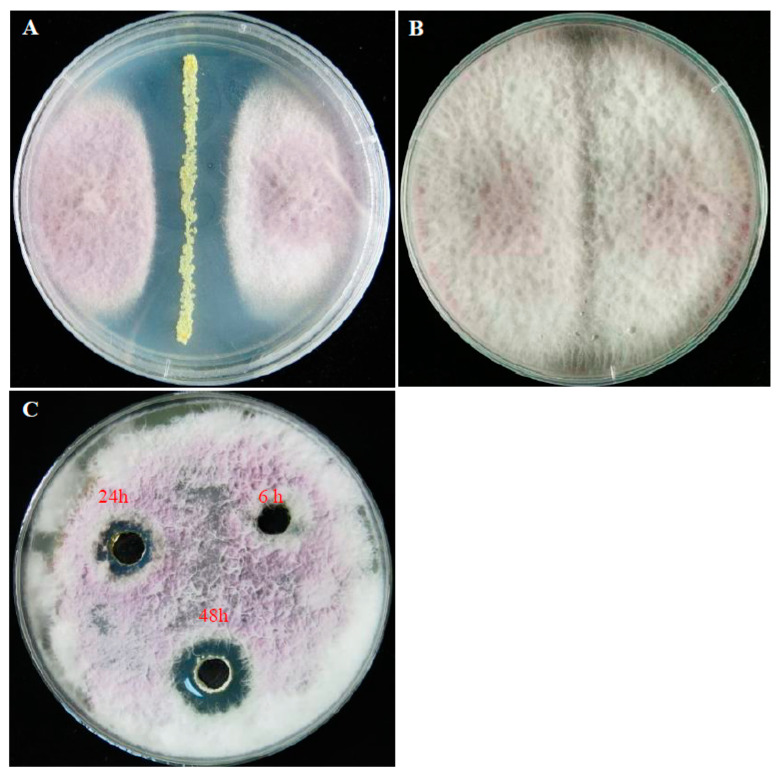
Inhibitory activity of *S. alfalfae* strain 11F. Compared with the control growth on PDA medium (**B**), *F. oxysporum* f. sp. *niveum* mycelial growth was inhibited by strain 11F (**A**). The strain 11F fermentation broth supernatant collected at different time points (6, 24, and 48 h) inhibited *F. oxysporum* f. sp. *niveum* mycelial growth. The inhibitory effect of strain 11F on mycelial growth is shown in (**C**).

**Figure 2 plants-12-03796-f002:**
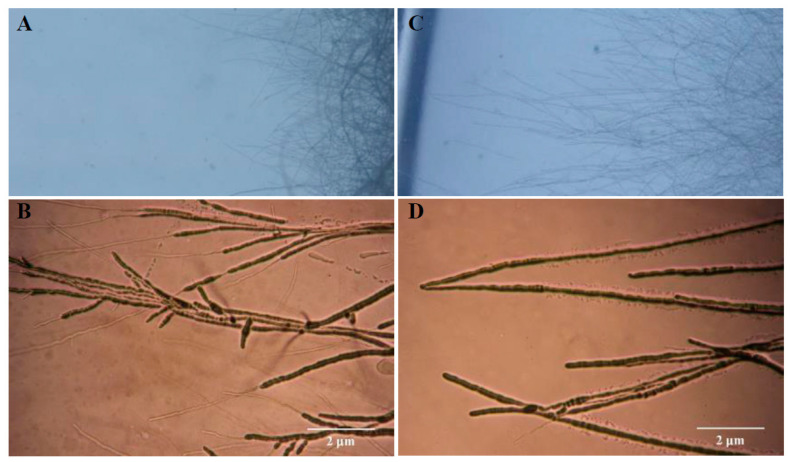
Effect of *S. alfalfae* strain 11F on *F. oxysporum* f. sp. *niveum* mycelial growth on PDA medium. Microscopic analysis of the morphological features of *F. oxysporum* f. sp. *niveum* mycelia treated with strain 11F (**A**,**B**) and the control mycelia (**C**,**D**).

**Figure 3 plants-12-03796-f003:**
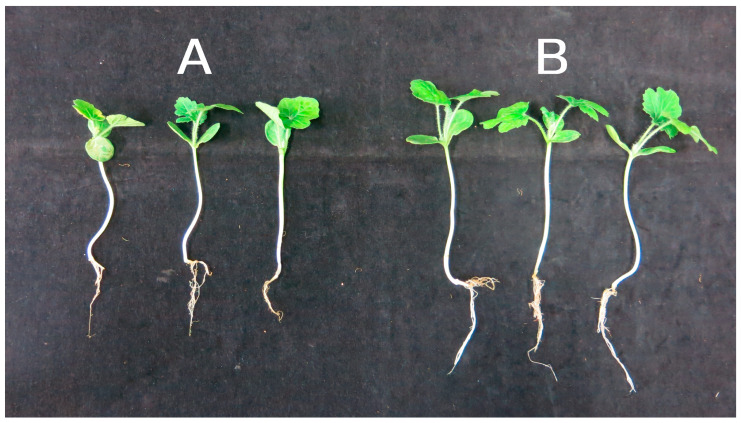
Effect of *S. alfalfae* strain 11F on watermelon seedling growth under greenhouse conditions. Left to right: watermelon seedlings treated with water (**A**) or 60-fold diluted fermentation broth (**B**).

**Figure 4 plants-12-03796-f004:**
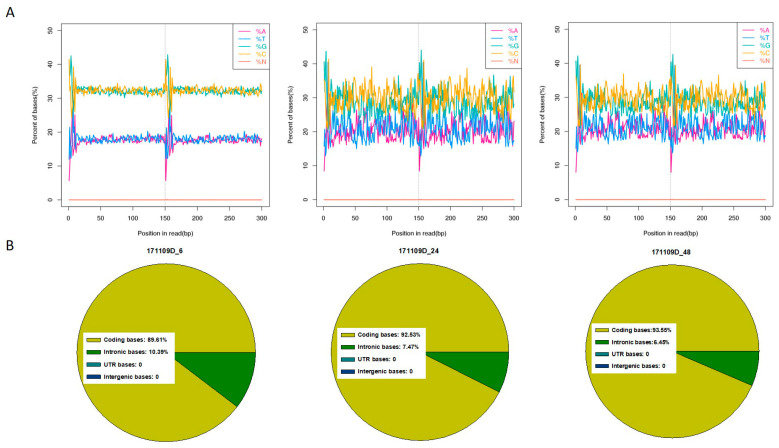
Transcriptome sequencing quality analysis of the samples at three fermentation time points (6, 24, and 48 h). (**A**) Base composition and quality distributions. (**B**) Distribution of gene coverage.

**Figure 5 plants-12-03796-f005:**
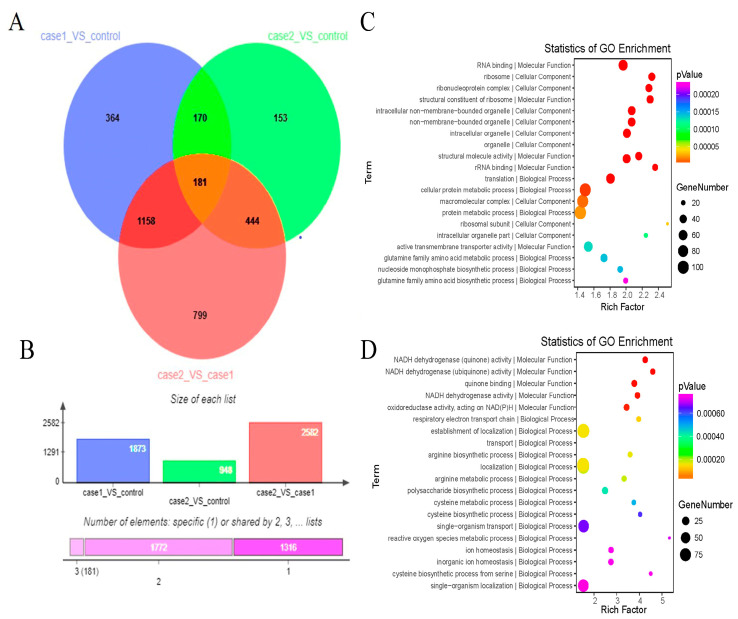
Analysis of differential gene expression among the three fermentation periods. (**A**) Venn diagram of differentially expressed genes (DEGs) common to the three comparisons. (**B**) Number of DEGs in each comparison. (**C**,**D**) Top 20 enriched GO terms. The Rich factor refers to the ratio of the number of differentially expressed transcripts in the GO entry to the total number of transcripts in the GO entry. An increase in the Rich factor reflects an increase in the degree of enrichment. The dot size indicates the number of DEGs assigned to that particular term, and the dot color indicates the *p*-value.

**Figure 6 plants-12-03796-f006:**
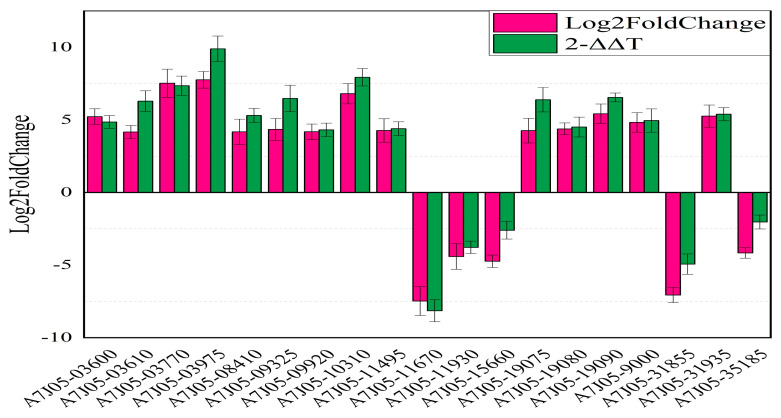
Relative expression levels of 19 differentially expressed genes verified by qRT-PCR analysis.

**Figure 7 plants-12-03796-f007:**
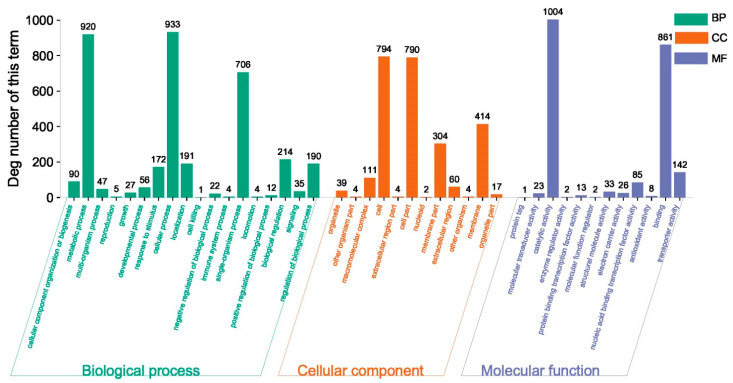
Distribution of GO categories (level 2) for *S. alfalfae* strain 11F transcripts. The GO functional annotations are summarized in three categories: cellular component, molecular function, and biological process.

**Figure 8 plants-12-03796-f008:**
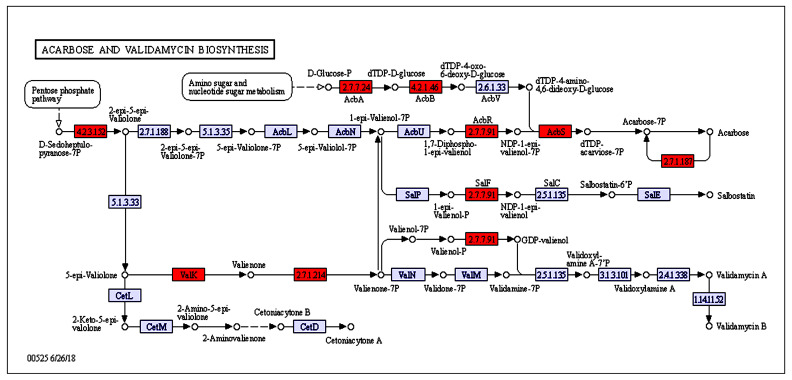
Schematic diagram of the acarbose and validamycin synthesis pathway. Red represents upregulated differential genes and violet represents downregulated differential genes.

**Figure 9 plants-12-03796-f009:**
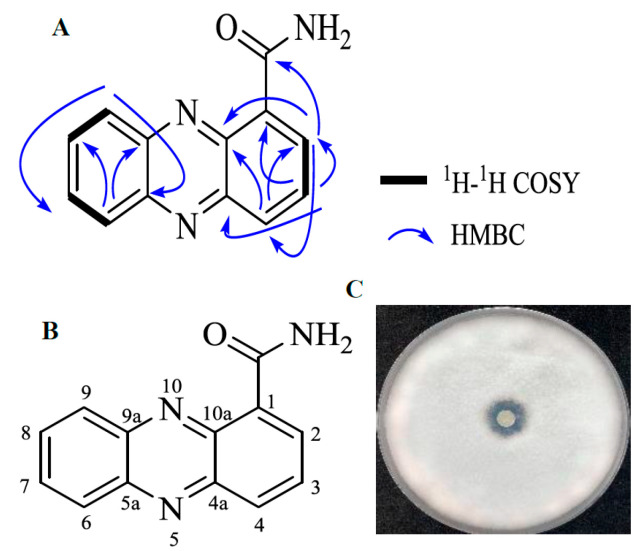
^1^H-^1^H COSY and key HMBC correlations of the isolated phenazine compound. (**A**) Structure of phenazine-1-carboxamide. (**B**) Phenazine-1-carboxamide purified from strain 11F fermentation broth. (**C**) Inhibition of *F. oxysporum* f. sp. *niveum* mycelial growth by the purified phenazine-1-carboxamide.

**Table 1 plants-12-03796-t001:** Effects of *S. alfalfae* strain 11F fermentation broth on watermelon seedling growth.

Treatment (*v*/*v*)	Plant Height/cm	Leaf Length	Leaf Width/cm	Fresh Weight/g	Dry Weight/g
Control	8.75 ± 0.18 b	2.90 ± 0.17 c	3.35 ± 0.11 c	0.82 ± 0.01 c	0.05 ± 0.00 c
11F (1:30)	9.20 ± 0.08 a	3.47 ± 0.13 b	3.45 ± 0.18 b	0.89 ± 0.19 b	0.07± 0.00 b
11F (1:60)	9.51 ± 0.13 a	4.08 ± 0.24 a	3.98 ± 0.19 a	1.01 ± 0.01 a	0.08 ± 0.01 a
11F (1:100)	8.32 ± 0.11 c	3.15 ± 0.03 b c	3.12 ± 0.10 b c	0.84 ± 0.02 c	0.06 ± 0.00 c

Note: different lowercase letters in each column indicate a significant difference (*p* < 0.05) and 1:30, 1:60, and 1:90 represent the volume ratios of fermentation broth (6 × 10^6^ cfu/mL) to water.

**Table 2 plants-12-03796-t002:** Effects of treatment with *S. alfalfae* strain 11F fermentation broth on incidence, disease index, and control effect of *Fusarium* wilt in watermelon seedlings.

Treatment	Incidence (%)	Disease Index	Control Effect (%)
Control	0.00 ± 0.00 c	0.00 ± 0.00 c	–
*F. oxysporum* f. sp. *niveum*	100.00 ± 0.00 a	73.59 ± 1.07 a	–
11F	0.00 ± 0.00 c	0.00 ± 0.00 c	–
*F. oxysporum* f. sp. *niveum* + 11F	25.14 ± 0.14 b	27.57 ± 0.18 b	62.54

Note: different lowercase letters in each column indicate a significant difference (*p* < 0.05).

**Table 3 plants-12-03796-t003:** ^1^H (500 MHz) and ^13^C NMR (125 MHz) data for the phenazine compound isolated from the *S. alfalfae* strain 11F fermentation broth (δ in ppm, J in Hz, in DMSO-d6).

No.	δH (J in Hz)	δC
1		131.0
2	8.69, dd (7.0, 1.7)	134.1
3	8.07, m	130.3
4	8.44, dd (8.7, 1.7)	133.0
4a		142.8
5a		141.3
6	8.42, m	129.3
7	8.06, m	132.0
8	8.04, m	131.6
9	8.31, m	129.2
9a		142.6
10a		140.2
CONH2	Ha: 9.73, s Hb: 8.09, br s	165.7

## Data Availability

The sequencing datasets generated in the current study have been submitted to the NCBI Sequence Read Archive (SRA; https://www.ncbi.nlm.nih.gov/sra, accessed on 17 October 2021) under accession number PRJNA771952 (https://www.ncbi.nlm.nih.gov/Traces/study/?acc=PRJNA771952, accessed on 17 October 2021).

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
