# Peer review of "Antagonistic Activity of Streptomyces alfalfae 11F against Fusarium Wilt of Watermelon and Transcriptome Analysis Provides Insights into the Synthesis of Phenazine-1-Carboxamide"

_plants, 2023, doi:10.3390/plants12223796_

Round 1

Reviewer 1 Report

This is a good-quality manuscript with properly planned methods and nicely presented results supporting the conclusions. The resolution of some figures should be improved. This research calls for the study of S. alfalfae metabolism study in response to pathogens. However, the amount of data is sufficient for the publication and I would be interested to read the follow-up publication. I have some minor comments which you can find below. Overall I consider this manuscript as acceptable for publication after minor revisions.

In-text comments:

Line 50: Please rewrite this sentence it is unclear.

Line 53: Remove: “and nutrients”

Line 56: Of what plant?

Line 62: One gene is considered not enough to assign strain to species level

Line 77: Please provide a table of results of antagonistic activity against all tested plant pathogens.

Line 79: Please provide a full species name when the organism is mentioned for the first time, please remember that abstract, main text, figure, and table captions are treated separately.

Line 80: Additional to what?

Line 87: to the morphology of control mycelium.

Line 104: P should be α

Line 109: Please define the disease severity index

Table 1: Please consider describing dilution as v:v (e.g. 1:30) ratio rather than how many times the medium was diluted.

Table 2: Please define: incidence, disease index, and control effect in the table caption. Additionally please give the name and the results of used statistical test. There are missing letters in the incidence and disease index comparison. You cannot compare statistically different measurements (each column should be treated separately).

Figure 3: Please add tags or frames to this figure.

Figure 4. Please provide a higher-resolution version of Figure 4A, and add the hours of fermentation to the figure.

Line 141: Please use hours of fermentation instead of Case 1 or 2 for clarity.

Figure 6: Please provide a higher-quality version of Figure 6.

Line 166: Please provide more detailed information for the selected genes.

Figure 5. Please add the gene-predicted function to the figure caption.

Line 207: Please exclude pathways for animals from the analysis of bacteria metabolism.

Figure 7: Please provide a higher-quality version of this figure.

Figure 8: Please extend the caption for this figure, explaining the colors and the program used to prepare the graphics. 

The language of the article is good with few minor issues. 

Author Response

Dear editor,

Thanks for your and the reviewers’ comments to our manuscript. We have revised the manuscript carefully according to the comments. All the corrections are labeled in red in the revised manuscript, and all the questions from the reviewers were answered and attached below.

I sincerely appreciate your review of our paper and I hope our changes to the paper are satisfactory.  Please contact me if you require additional information and thank you very much for handling of our manuscript.

Sincerely,

Huiling, Wu Ph.D,

Institute of Plant and Environment Protection, Beijing Academy of Agriculture and Forestry Sciences, Beijing 100097, China

Tel.: +86-10-51503337

Fax: +86-10-51503899

E-mail: wuhuiling925@126.com

Reviewer #1:

Line 50: Please rewrite this sentence it is unclear.

Answer: We have rewritten it.

Line 53: Remove: “and nutrients”

Answer: We have removed “and nutrients” in line 53.

Line 56: Of what plant?

Answer: We have changed it into “the soil near Qinghai Lake (101â—¦77′E, 36â—¦62′N) in China” in line 56.

Line 62: One gene is considered not enough to assign strain to species level

Answer: The strain assign to S. alfalfa according to 16S rRNA sequences and analyses of genomic collinearity.

Line 77: Please provide a table of results of antagonistic activity against all tested plant pathogens.

Answer: We have provided a table of results of antagonistic activity against all tested plant pathogens in supplementary material Table S1.

Line 79: Please provide a full species name when the organism is mentioned for the first time, please remember that abstract, main text, figure, and table captions are treated separately.

Answer: We have changed it into Fusarium oxysporum f. sp. niveum

Line 80: Additional to what?

Answer: We have deleted it.

Line 87: to the morphology of control mycelium.

Answer: We have revised it.

Line 104: P should be α

Answer: We have revised it and given the name and the results of used statistical test.

Line 109: Please define the disease severity index

Answer: We have defined the disease severity index in Materials and methods part 4.3.

Table 1: Please consider describing dilution as v:v (e.g. 1:30) ratio rather than how many times the medium was diluted.

Answer: We have revised it.

Table 2: Please define: incidence, disease index, and control effect in the table caption. Additionally please give the name and the results of used statistical test. There are missing letters in the incidence and disease index comparison. You cannot compare statistically different measurements (each column should be treated separately).

Answer: We have revised it and given the name and the results of used statistical test.

Figure 3: Please add tags or frames to this figure.

Answer: We have added it

Figure 4 Please provide a higher-resolution version of Figure 4A, and add the hours of fermentation to the figure.

Answer: We have provided a higher-resolution version of Figure 4A and added the hours of fermentation to the figure.

Line 141: Please use hours of fermentation instead of Case 1 or 2 for clarity.

Answer: We have revised it.

Figure 6: Please provide a higher-quality version of Figure 6.

Answer: We have provided a higher-resolution version of Figure 6.

Line 166: Please provide more detailed information for the selected genes.

Answer: The detailed information for the selected genes were provided in Supplementary material Table S4.

Figure 5: Please add the gene-predicted function to the figure caption.

Answer: The gene-predicted functions were provided in Supplementary material Table S4.

Line 207: Please exclude pathways for animals from the analysis of bacteria metabolism.

Answer: We have excluded pathways for animals from the analysis of bacteria metabolism.

Figure 7: Please provide a higher-quality version of this figure.

 Answer: We have provided a higher-resolution version of Figure 7.

Figure 8: Please extend the caption for this figure, explaining the colors and the program used to prepare the graphics. 

Answer: We have extended the caption for this figure, and have explained the colors and the program used to prepare the graphics.

Reviewer 2 Report

Dear Authors, the work presented to me for review is very valuable and has innovative research results. However, before publishing it, I recommend making corrections. I marked my comments on the manuscript of the work that I am enclosing to you. First of all, the introduction of the work and certain sentences in the discussion of the results should be improved. I also believe that the conclusions should be more specific. They refer again to the research results obtained, but are not specific. They need to be more precise and not repeat what you have obtained. You must answer the question of how the research results can be used in practice. What procedures should be implemented to be effective and economical?

Author Response

Reviewer #2:

Dear Authors, the work presented to me for review is very valuable and has innovative research results. However, before publishing it, I recommend making corrections. I marked my comments on the manuscript of the work that I am enclosing to you. First of all, the introduction of the work and certain sentences in the discussion of the results should be improved. I also believe that the conclusions should be more specific. They refer again to the research results obtained, but are not specific. They need to be more precise and not repeat what you have obtained. You must answer the question of how the research results can be used in practice. What procedures should be implemented to be effective and economical?

Answer: We have improved language of the introduction and discussion parts, and have specified the conclusions part. We have answered how the research results can be used in practice in the Discussion part.
